# Prototype of the Runway Monitoring Process at Smaller Airports: Edvard Rusjan Airport Maribor

**Boštjan Kovačič [1],\*[iD], Damjan Želodec [2] and Damjan Doler [3]**

[1]   Faculty of Civil Engineering, Transportation Engineering and Architecture, University of Maribor, Smetanova 17, 2000 Maribor, Slovenia

[2]   DRI Investment Management, Company for Development of Infrastructure Ltd., Kotnikova ulica 40, SI-1000 Ljubljana, Slovenia; damjan.zelodec@dri.si

[3]   Ministry of the Environment and Spatial Planning, Spatial Planning, Construction and Housing Directorate, Dunajska cesta 48, SI-1000 Ljubljana, Slovenia; damjan.doler1@gov.si

\*   Correspondence: bostjan.kovacic@um.si

**Abstract:** The last 20-year announcement predicts a 3.5% increase in the number of yearly passengers which will result in the doubling of the number of passengers in air transport by 2037. Such anticipation indicates the need for efficient monitoring of airport infrastructure as the support of opportune and efficient maintenance works. The novelties of this article are a process model of maintenance and monitoring, suitable for smaller and less burdened airports, and the methodology of monitoring of runways by implementation of the geodetic and geomechanics falling weight deflectometer (FWD) method. In addition, the results confirm the assumption that a specific environment such as an airport allows for sufficiently reliable determination of deformation areas or areas of vertical deviations of runways in a relative short time period available for measurements by using geodetic methods only or by combining other methods; our research model includes the FWD method. With the research, we have also shown there is an interaction between deformations or areas of vertical deviations on the surface and anomalies in the runway lower constructure which will, hereinafter, allow the development of the prediction, creating a vertical deviations or deformation model.

**Keywords:** geodesy; FWD; airport; deformations; vertical deviations; monitoring; geo-information model; measurements

## 1. Introduction

Today's society has a high demand and need for mobility. However, mobility is not a characteristic only of modern society; this need for movement dates back to the oldest civilizations. Orbanić and Rosi [1] claim that mobility is the key factor for social and economic development, allowing access to education, employment, health care, and markets; thus, it plays an important role in the elimination of inequality. Moreover, Orbanić and Rosi [1] indicate that, in addition to mobility, the main drivers of contemporary world processes include transportation and logistics as important carriers of globalization and development.

Increasing demand for travel to third countries and poorly logistically connected European areas can more than double the activities related to air traffic [2]. Generally, air traffic is efficient when it provides fast and, most importantly, safe transport of people and goods, which requires a modern and suitably maintained airport infrastructure—first of all, the runway. In particular, airport infrastructure is important in areas where the development of road and railway infrastructure is difficult due to the disadvantageous physical–geographical characteristics. Runways should provide adequate load capacity, good surface friction, and smooth aircraft running. All these requirements depend on the

proper maintenance of runways. Basically, well-maintained runways are important for an aircraft's safe take-off and landing [3]. Deformations on the surface distract the pilot's control of the aircraft and cause vibrations on the deck which obstruct proper reading of the measurement instruments, cause mechanical damage to the aircraft, and reduce the tire's contact with the ground, affecting the proper functioning of the braking system [4], damaging sensitive cargo, etc. The maintenance of runways should be carried out periodically, according to short-term, medium-term, and long-term plans, known in advance, using familiar equipment and methods in order maintain them as functional as possible. There should be as few interventions and traffic jams as possible to preserve runways in the best possible condition and in an economically favorable way. Late renovation of deformations causes them to appear to a larger extent, resulting in higher costs and longer periods of renovation. In addition, efficient airport management requires lots of data regarding the composition and condition of the airport's facilities, activities, and its surroundings. The data should be up to date, accurate, and accessible to airport staff at any given time. They can be obtained manually by the airport staff according to the prescribed protocols; however, this is a time-consuming activity and a possible source of human error. Moreover, data can be obtained and processed by completely automated or partially automated procedures and can be a part of the existing or developing geo-information systems. Requesting their updating, accuracy, and availability at any time encourages the development of systems that provide automated acquisition and processing in real time [5].

The downside of the existing models of monitoring the runway state is that they are intended especially for the analysis of the runway grip; the geodetic or other non-invasive methods which also allow detecting the deformations with high accuracy are not included, and due to their complexity, they are adequate only for bigger airports which have also, among other things, an adequate personnel cast. As demonstrated by D'Apuzzo et al. [4], accurate monitoring and prediction of runway situations are the main elements of developing models for the monitoring of runway conditions.

Detecting deformations and determining their shape, dimensions, and reasons for their occurrence is a complex process which requires an interdisciplinary approach [6,7]. Typically, an interdisciplinary approach is used in many modern scientific studies because the exposed issues should be explored from several aspects. In the last period, automated gathering and processing of spatial data regarding deformations on pavement structures, as well as on runways, are the subject of many studies [8–15]. Their research mostly focuses on interpretation and analysis of the pavement and runway snapshots and development of algorithms for automated deformation recognition or extraction (cracks, holes, etc.) from snapshots, determination of deformation dimensions (i.e., width and depth), and their classification.

In general, many authors have researched the usage of various methods for the condition analysis of pavement structures. As stated by Doler and Kovačič [16], the authors studied the implementation of 3D laser scanning, ground penetrating radar (GPR), falling weight deflectometer (FWD), and SAR interferometry [17–35]. The analysis of the conditions of the pavement structure is generally carried out using non-destructive methods. Among them, FWD [36–41] is the most widely used. The results of these analyses represent one of the bases for the development of such pavement structures that will withstand increasing loads. However, the accessible literature does not provide studies where FWD is combined with geodetic methods used for condition analyses on the surface of pavement structures and, among them, conditions on the runway surfaces.

Our research goals were to provide a process model for maintenance and monitoring, suitable for smaller and less burdened airports, and to test the methodology of runway monitoring. For the needs of the research, we have set forth goals, based on the above-described issue, and we set the working hypothesis: "The monitoring model which was developed by Doler and Kovačič [16] allows detailed consideration of deformations or vertical deviations and causes for their creation." This research also provides answers on two research questions: "Are the areas of the vertical deviation on the surface causally related to anomalies in the lower runway structure?" and "Can we predict the creation and expansion of the deformations or vertical deviations?"

Fundamentally, our contribution discusses the maintenance process model and the monitoring sub-process, intended for smaller and less burdened airports such as Edvard Rusjan Airport in Maribor, as well as a methodology for runway monitoring using a geodetic and geo-mechanical FWD method. The suggested sub-process of runway monitoring is a novelty which allows the capture and processing of various types of data and provides the damage degree of pavement structures (i.e., runways) based on the analysis. The main added value of this research to the scientific community is the realization that there is an interaction between deformations or vertical deviation areas on the surface and anomalies in the lower runway structure which was shown with the combination of geodetic and FWD measurements, executed in the same coordinate system.

## 2. Equipment and Methods

Monitoring was performed at the Edvard Rusjan Airport in Maribor, which is the second largest international airport in Slovenia, shown in Figure 1. The airport has a specific environment with restricted movement, conducted in accordance with rules implemented at the airport. Therefore, a previous agreement with airport management was made in order to carry out the observations smoothly and during limited operations, which was before daily opening until 8.30 in the wintertime and until 7.30 in the summertime and between individual announcements of landings and take-offs.

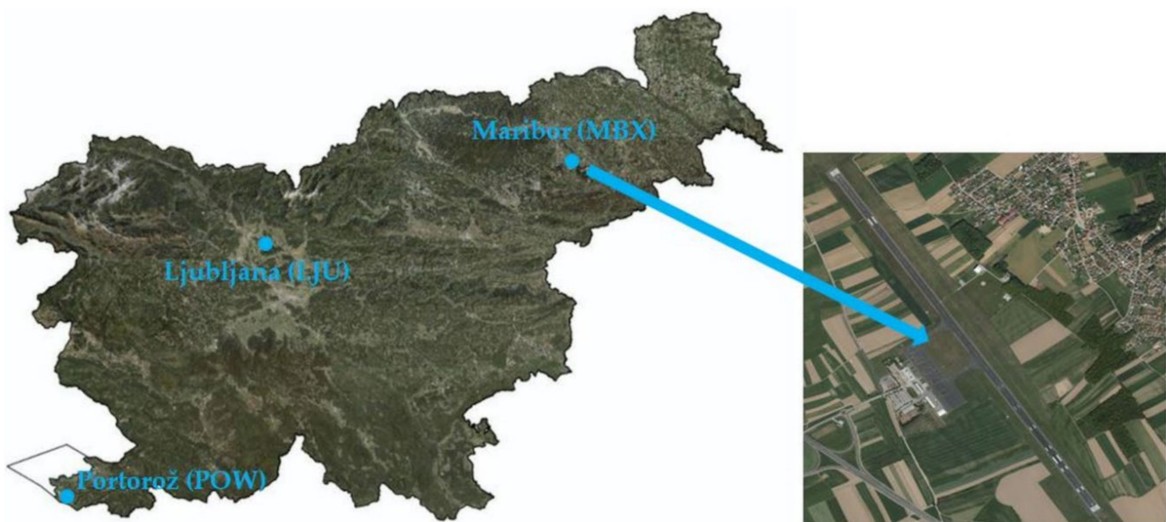

**Figure 1.** Location of Edvard Rusjan Airport in Maribor. Source: Administration of the Republic of Slovenia for Civil Protection and Disaster Relief and Surveying and Mapping Authority.

### 2.1. Process Model for Maintenance and Monitoring

Before performing field measurements, the process model was made to serve as a basis for the establishment of the airport pavement management system (APMS). Figure 2 shows the process structure for the APMS at the Edvard Rusjan Airport in Maribor (APMS-MB). Doler and Kovačič [16] developed a decision model aimed at runway monitoring (IMV-P) that predicts inclusion and analysis of several types of measurements. They proved that vertical deviations on runways can be determined on the basis of geodetic methods. Figures 3–5 include the IMV-P_all process model, which is an upgrade of the model developed by Doler and Kovačič [16]. In addition to geodetic measurements, the IMV-P_all model includes supplementary measurements.

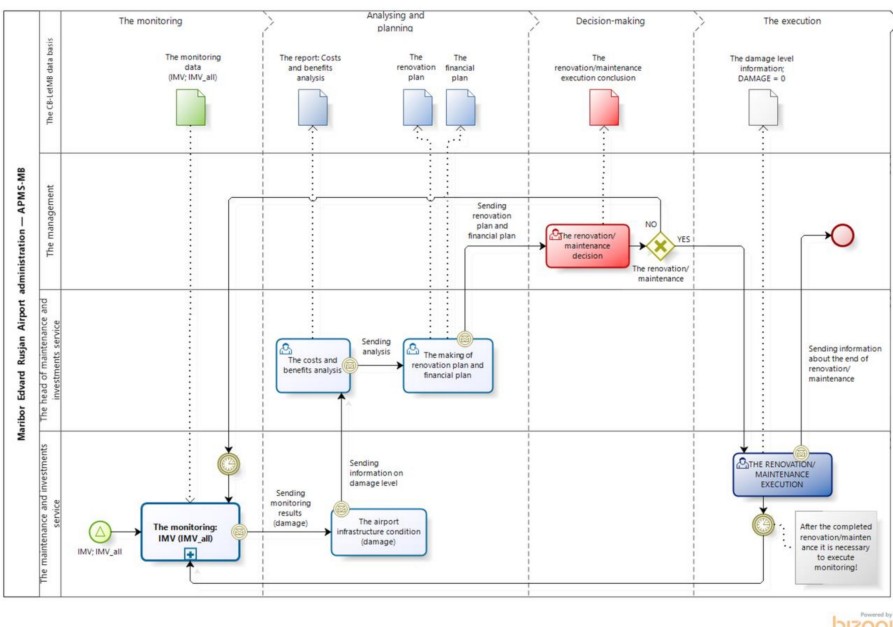

**Figure 2.** The APMS-MB, Edvard Rusjan Airport in Maribor (MB) process model.

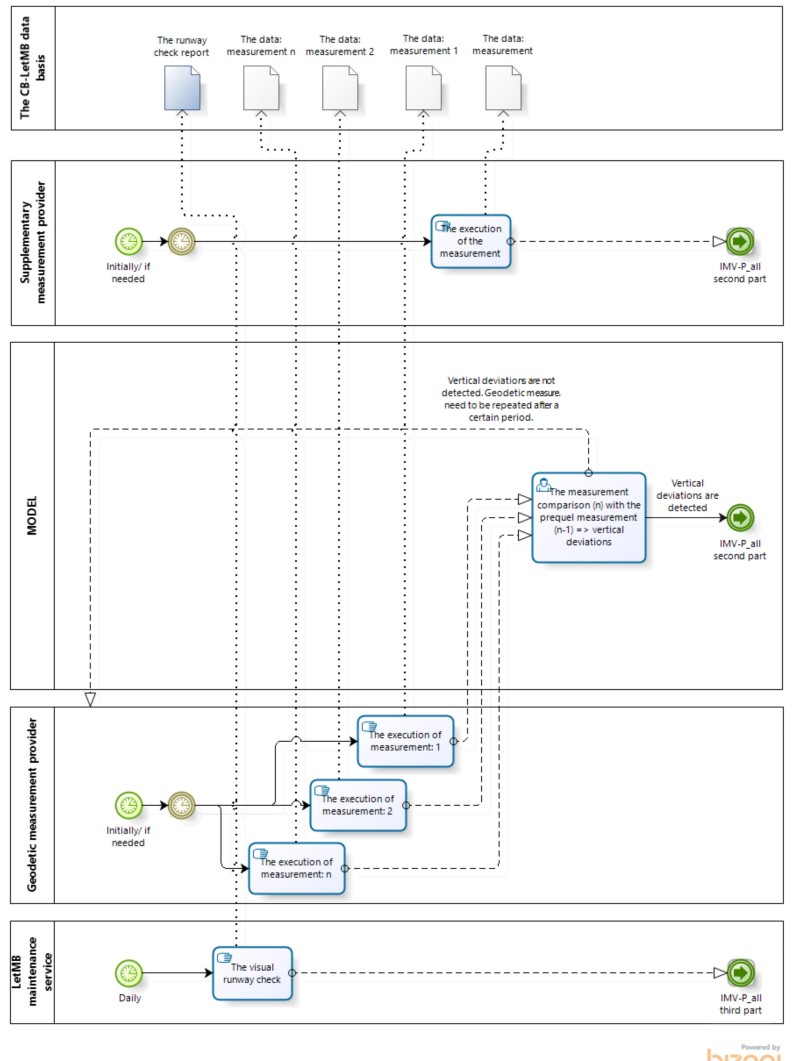

**Figure 3.** The monitoring process model, sub-process IMV-P_all, part 1.

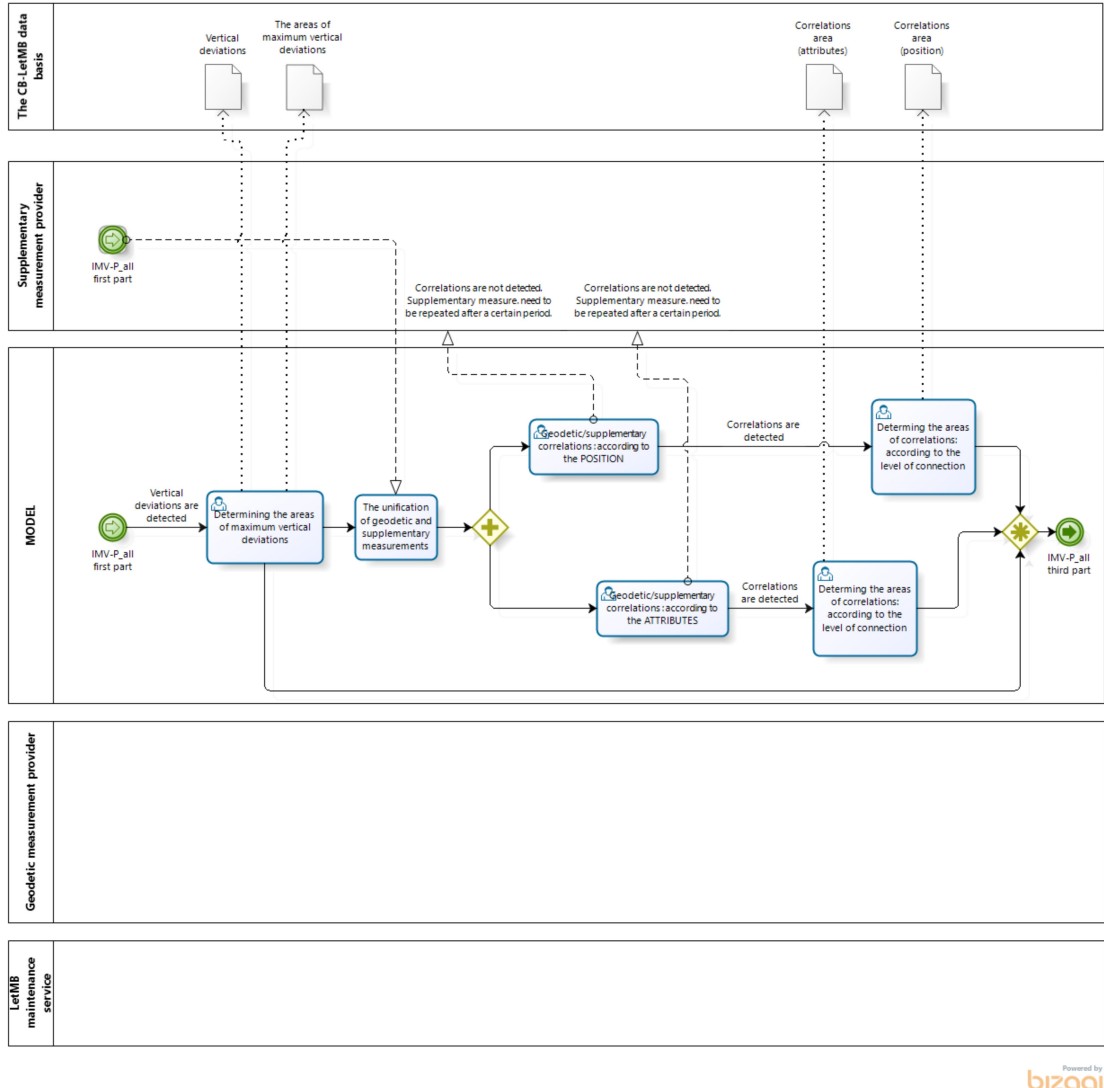

**Figure 4.** The monitoring process model, sub-process IMV-P_all, part 2.

Figures 3–5 show that the IMV-P monitoring model was expanded in order to include new actors and anticipate the involvement of a supplementary measurement provider. The geodetic measurement provider and the supplementary measurement provider are included in the expanded IMV P_all monitoring process model as external actors. Thus, the result of the IMV-P_all monitoring model is specific areas of vertical displacements which are a combination of findings based on virtual runway inspection, results of the used geodetic method, and results of the supplementary method; obviously, our research included FWD measurements.

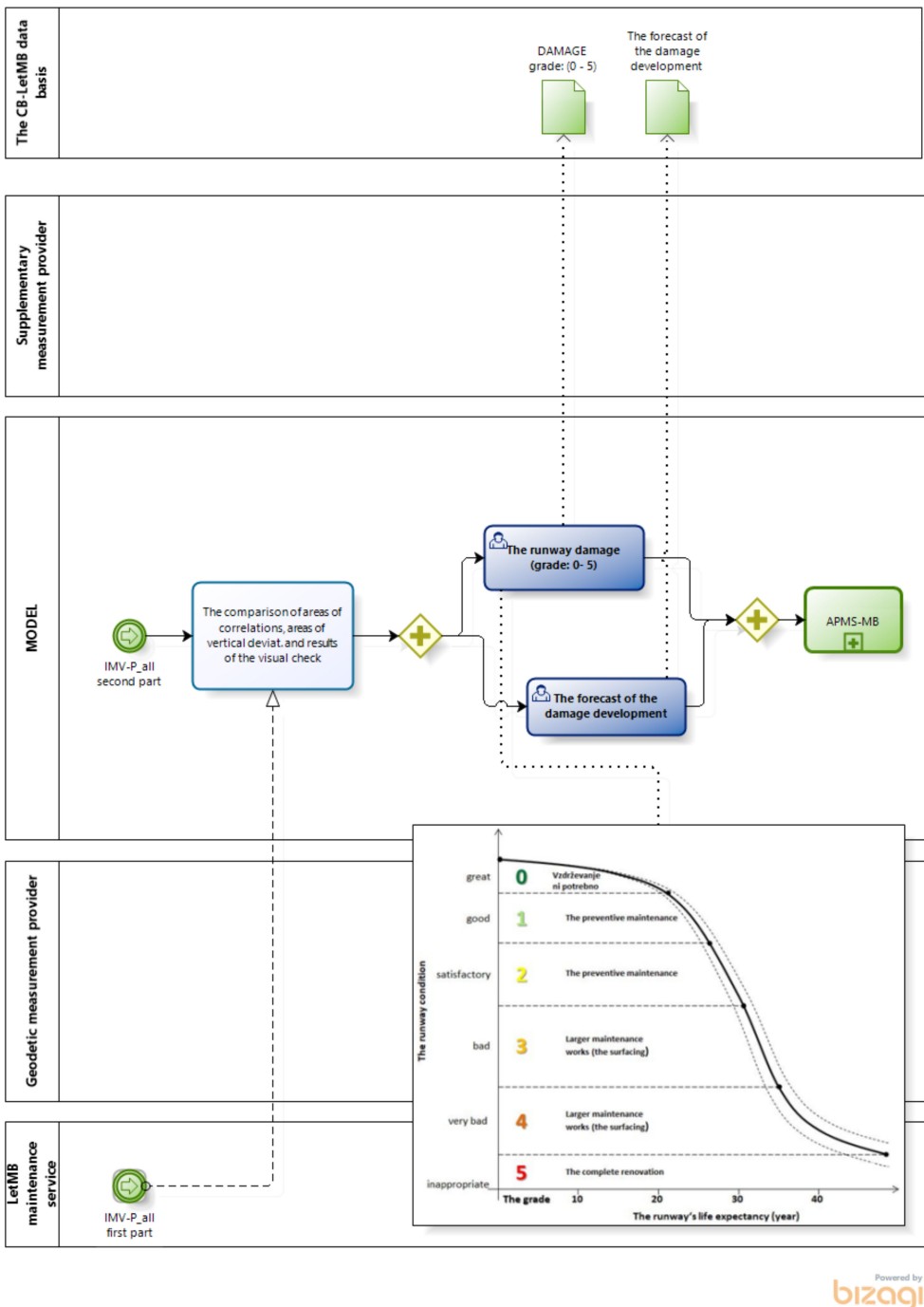

**Figure 5.** The monitoring process model, sub-process IMV-P_all, part 3.

## 2.2. Monitoring Methodology

Measurement was taken on the runway section called the touchdown zone, where the largest deformations are expected due to the high pressure from aircraft landings. The measurement area is 45 m wide and 300 m long (Figure 6).

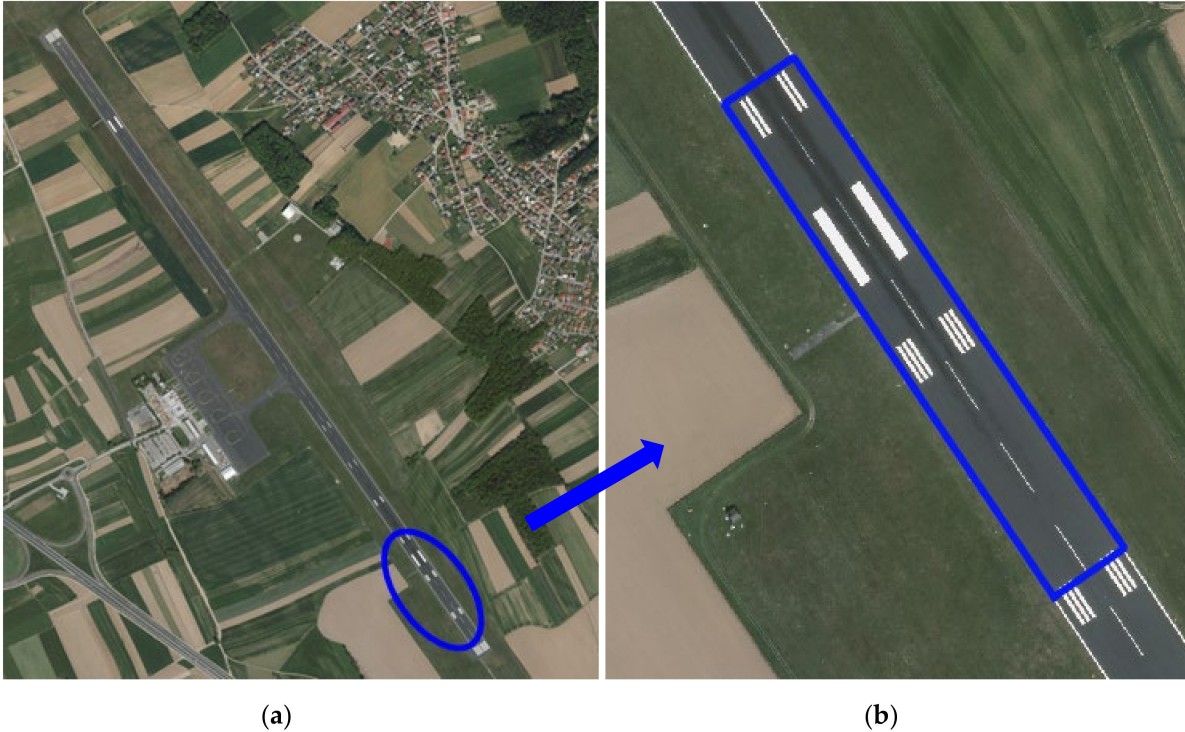

(**a**)　　　　　　　　　　　　　　　　　　　　　　(**b**)

**Figure 6.** The runway: (**a**) the measurement area; (**b**) the touchdown zone.

2.2.1. Used Measurement Equipment

A personal vehicle, unsprung trailer, robotic Leica TS50 total station, 360° total station prism Topcon A7, GNSS receiver Topcon HiperPro, and FWD measurement equipment were employed for measurements. The used equipment is shown in Figures 7 and 8.

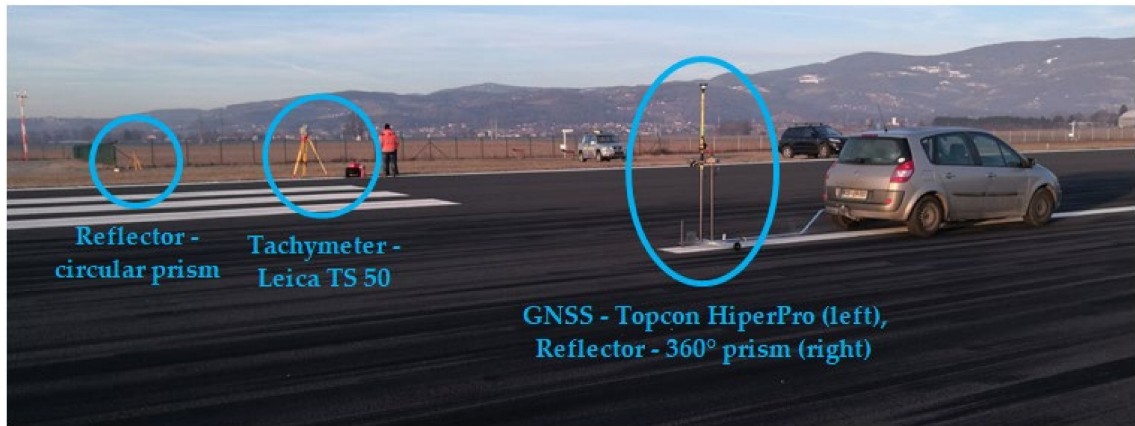

**Figure 7.** The used geodetic measurement equipment.

Moreover, the ductility measurements and analysis of the runway's load-bearing capacity were performed in parallel with the measurements of the vertical deviations formed on the runway's upper surface. The ductility measurements were carried out by FWD measurements that measure the ductility of the pavement's structure. In order to perform FWD measurements, the following were used: a tow car with a laptop, a trailer with FWD measurement equipment from Dynatest (8012 (FastFWD) model), a GNSS receiver, and equipment to take the temperature of the asphalt layers (i.e., drilling machine and thermometer) (Figure 8).

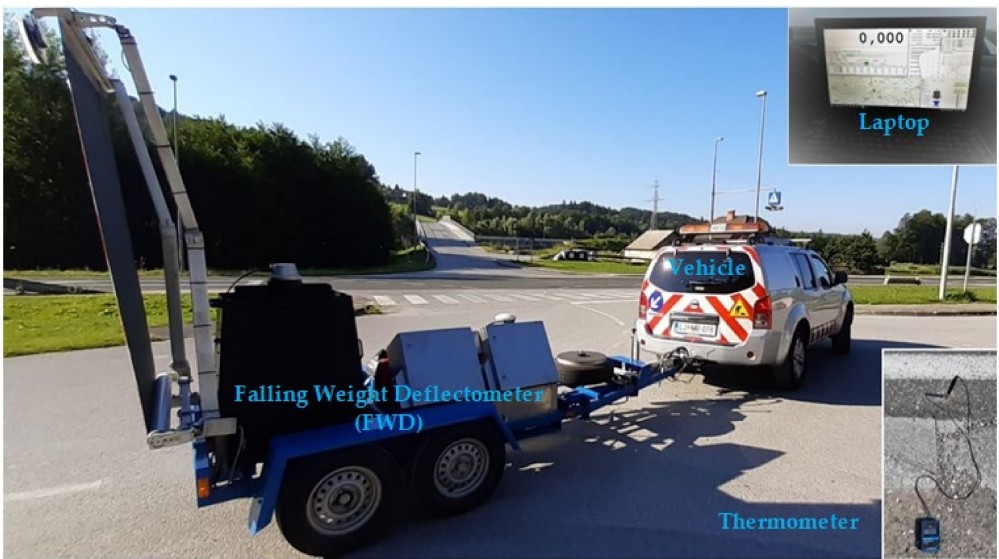

**Figure 8.** The used falling weight deflectometer (FWD) measurement equipment.

The tow car was employed to move the FWD equipment to the place of measurement. A laptop (Figure 9) in the vehicle cab was installed with the software DDC, "Dynatest data collection software", which aims to measure the data captured and save them, while synchronizing various data measurements, in our case location, GNSS measurement data, and FWD measurement data. At the same time, it allows the operator to monitor and control the measurement performance from the vehicle.

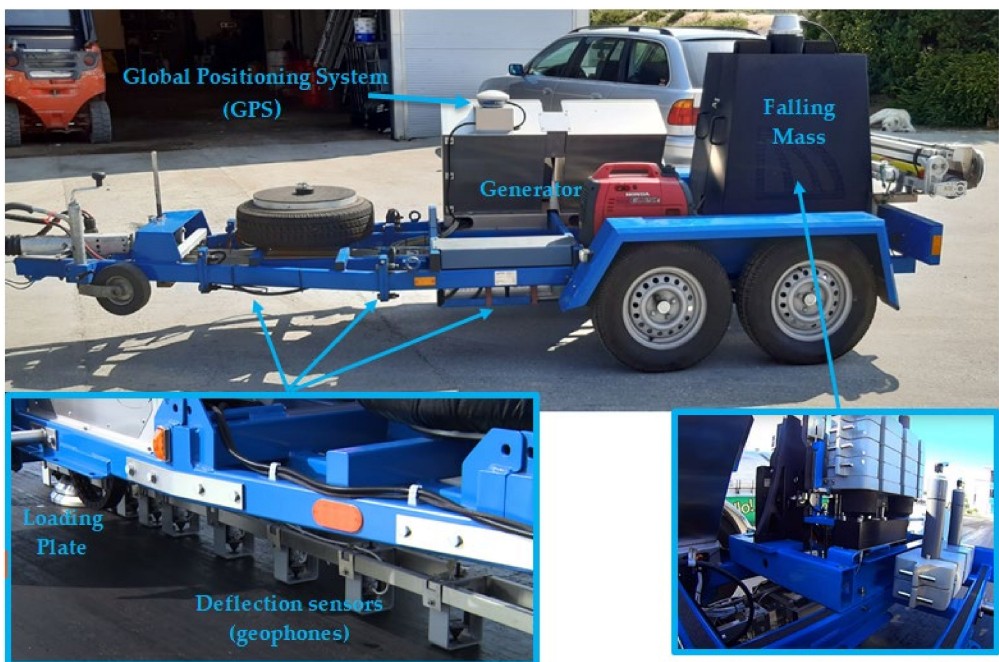

**Figure 9.** The trailer with the FWD measuring equipment.

The measurement equipment, which was installed on a two-axle trailer, included three basic parts (Figure 9): FWD measurement equipment; a GNSS receiver that allowed for the collection of location data; and an electric generator that provided the necessary electricity. The central part was the measurement equipment which had components including a loading plate, falling weights, and ductility/deflection meters—geophones (Figure 9).

### 2.2.2. Methodology for Taking Geodetic Measurements

The geodetic measurements were carried out in two phases. A basic geodetic grid was developed and measured in the first phase that materializes a stable and homogeneous coordinate system and perpendicular grid. Grids were determined to serve as the starting point in the comparative analyses of the first, second, and third measurements and the calculation of the vertical deviation between them. The observations in the first phase were performed on 27 December 2017. In the second phase, observations were conducted via the trailer, specially designed for this type of measurement. The trailer was without suspension in order to detect irregularities—deformations—better. It was equipped with measurement sensors (the 360° prism, the GNSS receiver, self-adhesive reflective targets, etc.). The second phase was performed over three periods of measurements: the first during a period of low temperatures (25 January 2018), the second over a period of medium temperatures (5 April 2018), and the third during a period of higher temperatures (8 June 2018). Namely, intensive vertical displacements were expected on the runway in these time periods due to the freezing and thawing of the ground under the runway. We needed 2 h and 30 min in the first phase for the execution of the geodetic measurements in the dimension area. In the second phase, three time dimensions were executed. We needed 30 min for each time measurement.

Measurements with the robotic Leica TS50 total station were carried out from the standing point 20002 (Figure 10). Coordinate standing points and orientation points were determined in the corresponding coordinate system with the GNSS system. The base station of the GNSS receiver was placed on the 20001 point, which was used in further measurements as an orientation point. Importantly, the points of the basic geodetic grid were re-measured and leveled before each time measurement. In fact, deformation analysis according to the Hannover method [42] was used in our case. The points of the perpendicular grid provided for the determination of the comparison plane, calculated according to the Moor–Penrose method [43]. The points of the geodetic grid and the points of the measured perpendicular grid (approximately 6 × 6 m) are shown in Figure 10.

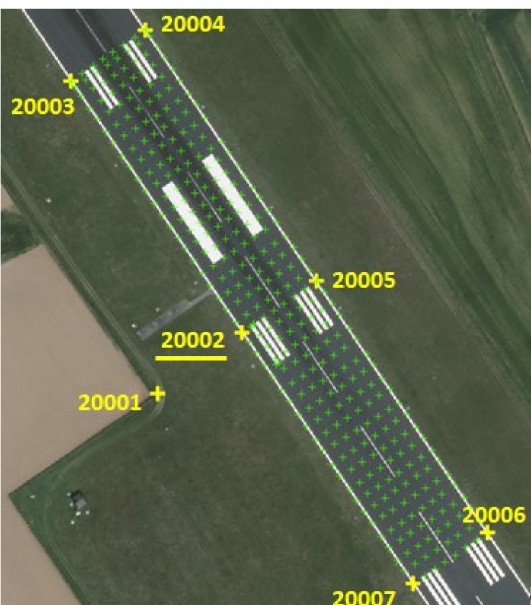

**Figure 10.** The basic geodetic grid (yellow crosses) and measured perpendicular grid (green crosses).

The measurement with the trailer was performed in the second phase over three measurement periods. The trailer was driven by the car at a constant speed of 6 km/h. The movement speed was the minimum speed achieved by the car. The 360° reflective prism and GNSS receiver mobile station were installed on the trailer. In addition, the prism positions were monitored during its movement by the robotic Leica TS50 total station with a frequency of 10 readings per second. The first measurements

provided the positions of 8607 points, the second measurement provided the positions of 7822 points, and the third measurement provided 7612 points over the entire trajectory of the trailer. However, only the points of the trajectory within the measurement area were taken for further processing (the first measurement: 6940 points, the second: 6780 points, and the third: 6466 points), i.e., without the points gained from the trailer turn. The regression line was determined for each period measurement and for each trajectory line as well as its deviations (upwards and downwards). Five areas were determined where the deviations were larger than ±8 cm, which is the maximum deformation according to the airport's regulations on runway maintenance. The comparison of the results shows that the areas of the first, second, and third period measurements coincided (Figure 11). However, larger deviations were detected in areas 3 and 4, which resulted in further comparison of the time measurements on the joint area of areas 3 and 4 (Figure 12).

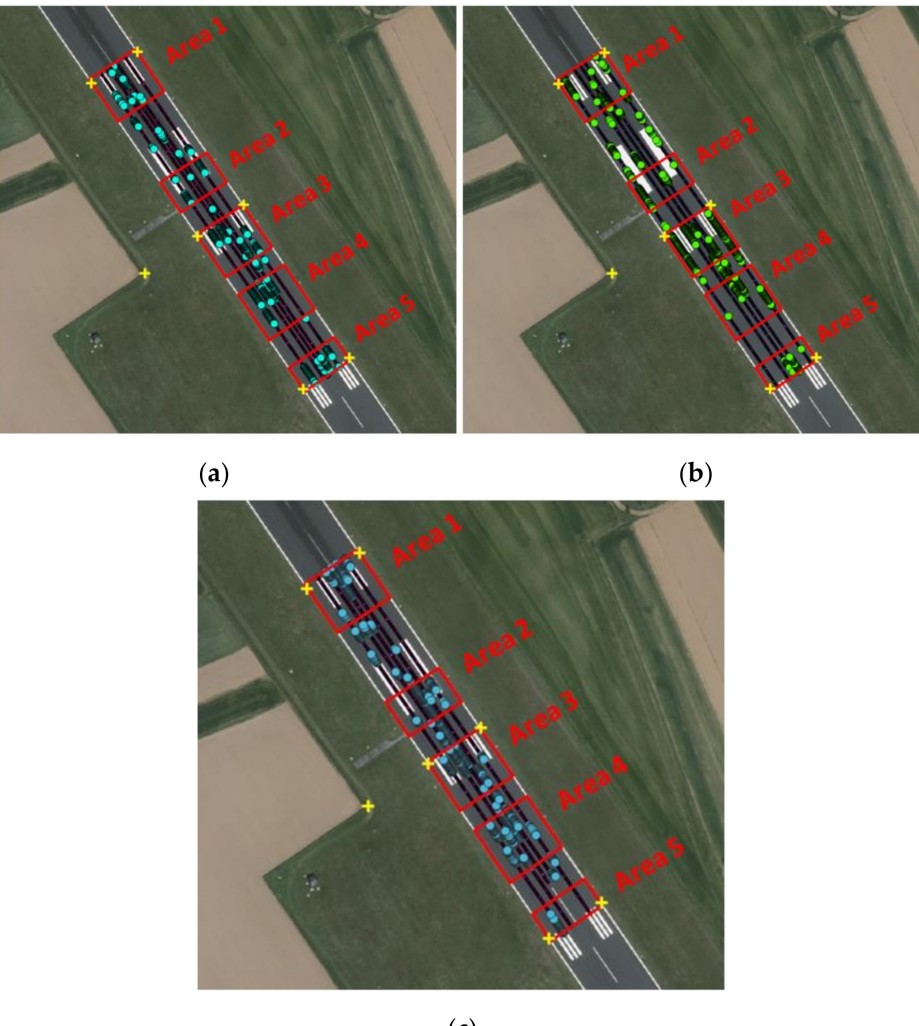

**Figure 11.** Areas of maximum vertical deviations in the (**a**) first, (**b**) second, and (**c**) third periods of measurement.

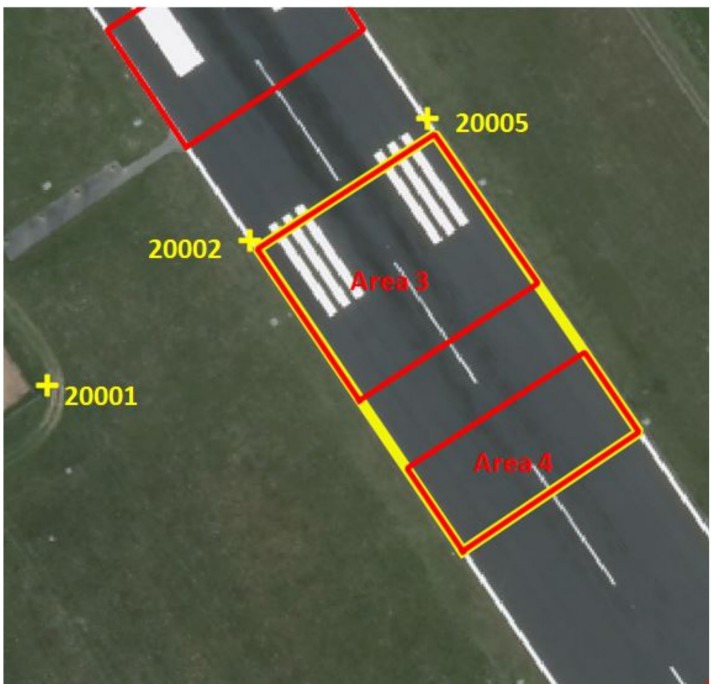

**Figure 12.** Comparison area.

### 2.2.3. Methodology for FWD Measurement Execution

Generally, the methodology of executing FWD measurements is simple and forecasts one phase only. We needed 1 h and 10 min for the execution of the FWD measurements in the dimension area. The car tows the measurement trailer to the measurement area. The measurements are taken on the spots where the prequel geodetic method senses the areas with larger vertical deviations. The operator triggers the measurement from the car cab which results in lowering the falling weight from a certain height to the loading plate and pushing it perpendicularly to the pavement structure which deflects due to the force application. The weight falls from such a height that its free fall causes a wheel load of 50 kN. The wheel load is equal to half of the axle load. In our case, the fall under the loading plate with an area of 0.0707 $m^2$ resulted in tension of 707 kN/$m^2$. The frequency of executed loads and the form of the loading plate were determined by the simulated load of the pavement structure with the force of 50 kN applied by the truck or the load of 120 kN in our case, which is generally caused by an aircraft in the middle range such as Airbus A320 and Boeing 737 mp. Different loads give different deflections and show, at the same time, the level of pavement structure deflection. First, three blows were carried out, then the measured value of the third blow was used [44].

Basically, deflection, which means bending, was implemented as the measure for the load of the pavement structures, and it is in inverse proportion to the load capacity. Deflection of the pavement structure is detected by special measurement sensors—geophones. These are differently remote from the loading plate and register deflections of the carriageway structure [45]. The first geophone was installed in the center of the loading plate; the second was at the distance of the diameter of the loading plate, measured from the center of the loading plate; and the third, fourth, and fifth were installed at distances larger than the equivalent thickness of the pavement's structure and smaller than the equivalent thickness of the pavement's structure increased by one meter. Normally, the standard installations of geophones in Slovenia, according to the center of the loading plate, are: 0 mm (D0), 200 mm (D200), 300 mm (D300), 450 mm (D450), 600 mm (D600), 900 mm (D900), 1200 mm (D1200), 1500 mm (D1500), and 1800 mm (D1800) (Figure 13) [44].

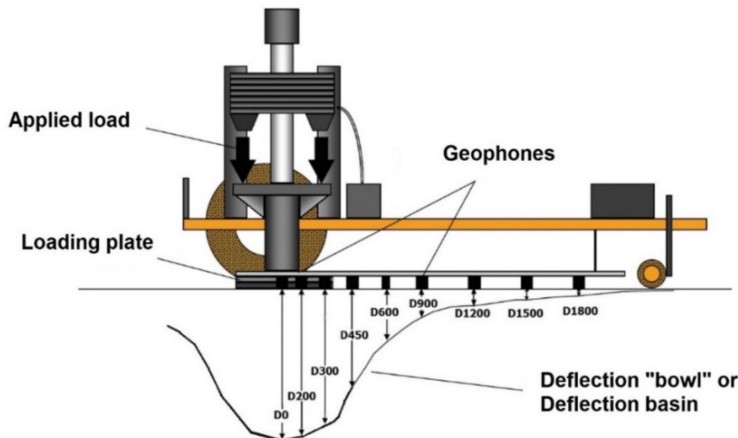

**Figure 13.** Presentation of the measurement with the FWD device [46].

The deflection measurements were performed on three axes: a right axis, a middle axis, and a left axis (Figure 14). The measurements aimed to detect the deflections on the runway at 120 kN loads on each wheel.

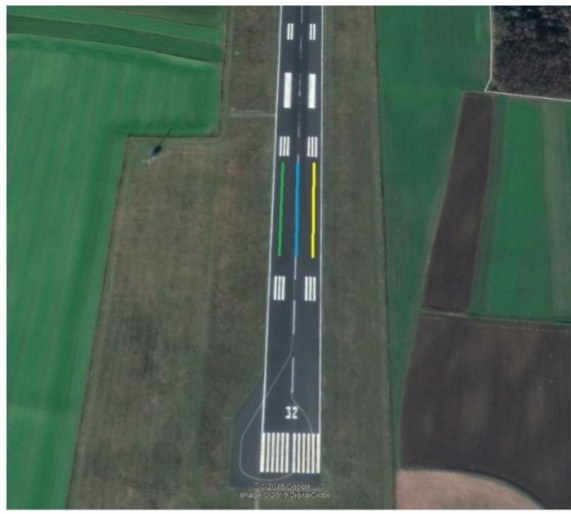

**Figure 14.** The measurement area: measurements axes—a right axis (green), a middle axis (blue), and a left axis (yellow).

FWD measurement analysis can be executed on the basis of the calculation of the elasticity modules, on the basis of the structural capacity of the pavement's structure according to the main deflection, or on the basis of the calculation of the deflection indices.

The FWD measurement analysis in our research was executed by the approach based on the evaluation of the structural capacity of the pavement structure, determined in respect to the deflection D0. The simplest method for the estimation of the structural capacity is evaluation and comparison of temperature-corrected main deflections (D0) [47]. The main deflection is the deflection directly under the load area. Among different methods of estimation of structural capacity, this method is the simplest and also the most robust because it demands only a minimal number of input data. However, it does not take into account the properties of the runway and it is also less accurate, especially on pavement structures with thicker load layers [44].

2.2.4. The Requirements of the Measurements

The monitoring results are determined in a stable and homogeneous coordinate system which presents a geometric base. On the dimension area, we materialized the coordinate system with the

basic grid net. The establishment methodology and the leveling are presented and described in detail in [48]. The average assessment of the height accuracy value ($\sigma_H$) after the leveling is $\sigma_H$ = 3.00 mm. The average assessment of the horizontal position accuracy (east—$\sigma_E$, north—$\sigma_N$) value after the leveling is $\sigma_E$ = 0.48 mm and $\sigma_N$ = 0.14 mm.

The stable geometric base allowed for us to achieve the demanded accuracy of determining the deformations on built objects, in our case the monitoring results, described in [49,50]. The average accuracy of the monitoring results ($\sigma_M$) is $\sigma_M$ = 0.49 mm.

## 3. Measurement Results

### 3.1. Results of Monitoring with Geodetic Measurements

Use of geodetic methods to determine vertical deviations was studied by Doler and Kovačič [16]. They leveled the results of geodetic measurements with the help of deformation analysis using the Hannover method [42] and calculated deviation planes with the help of the Moor–Penrose method [43]. The key result of the geodetic measures was the determination of areas of vertical deviations. Figure 15 shows vertical deviations in respect to the grid for the first measurement period. Bumps are indicated with the red color and hollows with the blue color.

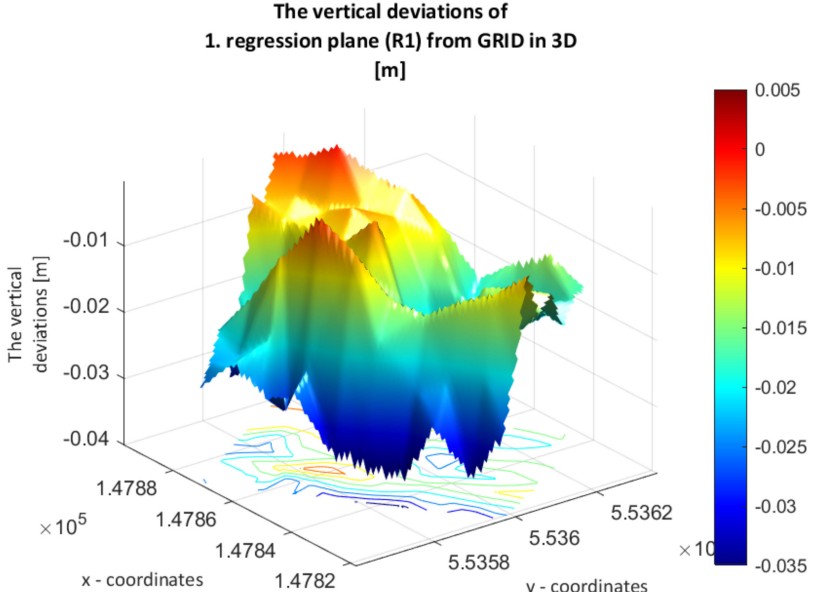

**Figure 15.** Vertical deviation of plane R1 from the grid.

### 3.2. Results of Monitoring with FWD Measurements

The analysis of the FWD measurements included deflection measurements on the middle axis, where the measured deflections were the largest. Figure 16 shows deflections on the middle axis. The measurements were executed at a 2 m distance from each other. The largest measured deflections were on the spot where the runway load was the highest. In addition, the deflection measurements were executed at 120 kN load on a wheel with an axis load of 240 kN. This simulated the landing loads of aircrafts such as the Airbus A320 and Boeing 737. The green line in Figure 16 shows the average deflection (534 μm). The values of deflections were above 450 μm up to the stationary 0.078 km, which indicated the decreased load capacity in this part of the runway. On the other hand, deflections between stationary 0.080 and 0.120 km had lower values and suggested better load capacity in this part of the runway.

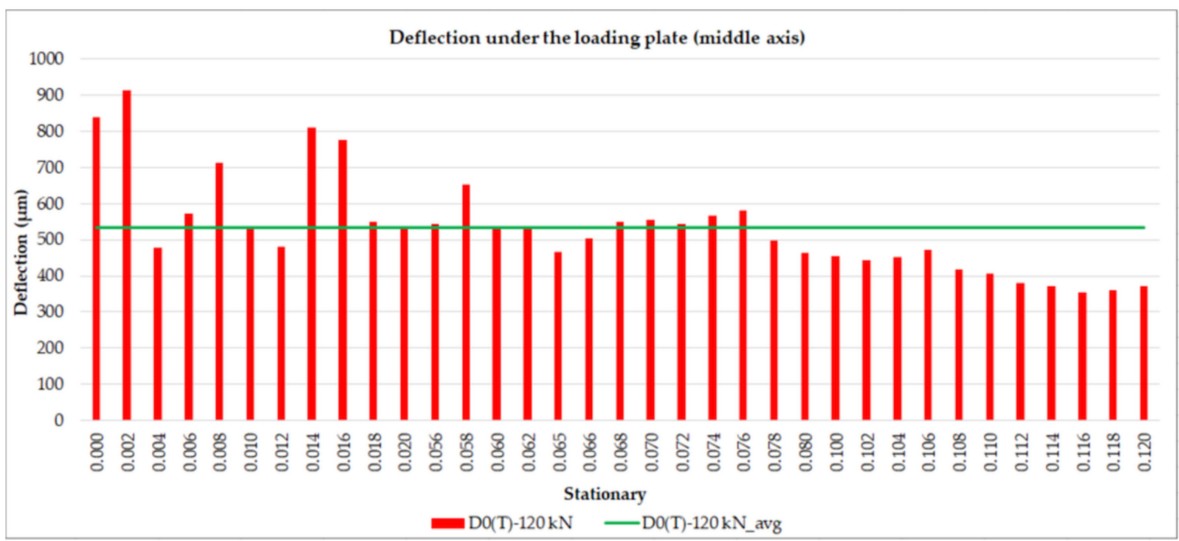

**Figure 16.** Temperature-corrected deflections D0(T) under the loading plate measured in the middle axis.

A demanded ground load capacity was prescribed for each airport, which is expressed by the Californian Bearing Ratio (CBR). The CBR index is determined according to deflections measured under the loading plate. Edvard Rusjan Airport in Maribor has the prescribed minimal value of CBR pavement—15% of Pavement Classification Number (PCN) 86. The average value of the CBR index in the middle axis amounted to 30%, which satisfied the required conditions.

### 3.3. Results of the Monitoring Using Geodetic and FWD Measurements

The expanded model of maintenance, narrower monitoring model anticipated the inclusion of supplementary measurements. However, one condition should be met, namely, supplementary measurements should be executed using the same coordinate system as the geodetic measurements. Figure 17 shows a combination of the results obtained by geodetic measurements and FWD measurements. In order to provide a comparison, suitable results were gained in the middle axis. The comparison shows that the vertical deviations on the runway's surface coincided with the measured deflections in terms of location. A schematic presentation of coincidence by local deformations (i.e., vertical deviations and anomalies) is displayed in Figures 17 and 18. Typically, FWD measurements give larger deflections in the areas where geodetic measurements provide larger vertical deviations (bumps and hollows). In Figure 17, three areas with red, black, and blue circle parts are marked with numbers 1, 2, and 3 which coincide with the areas of larger vertical deviations detected with the use of geodetic measurements and areas where with the FWD measurements' bigger deflection values were measured. Bigger deflection values are in Figure 17, marked on the graph with squares and numbers 1, 2, and 3. Importantly, the focus is only on the comparison of extreme values.

The comparison of the results of both measurements suggests the occurrence of a causal connection between the perceived vertical deviations that occur on the runway surface and anomalies in the runway underground. As a result, this finding provides further model development, which also enables the forecast of the appearance of vertical deviations. Figure 18 shows a schematic display of the locational connection of the vertical deviation areas on the surface which are detected with geodetic measurements and anomalies in the lower structure detected with the FWD measurements.

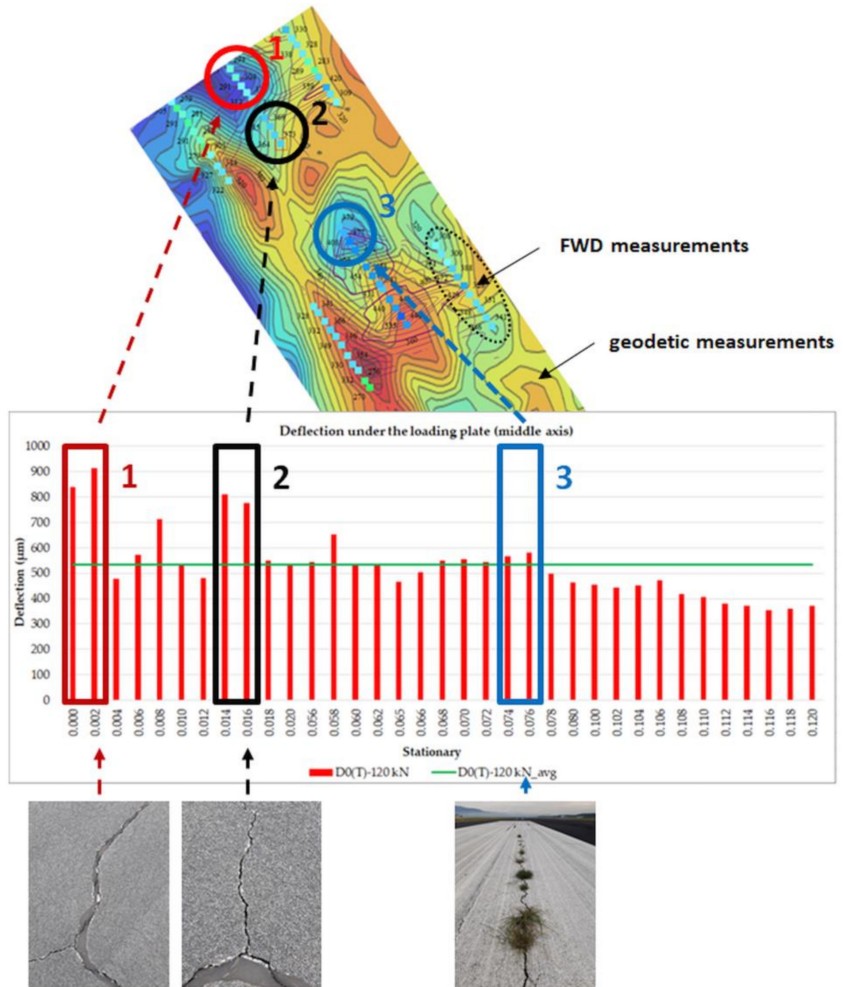

**Figure 17.** Comparison of the results gained by geodetic and FWD measurements.

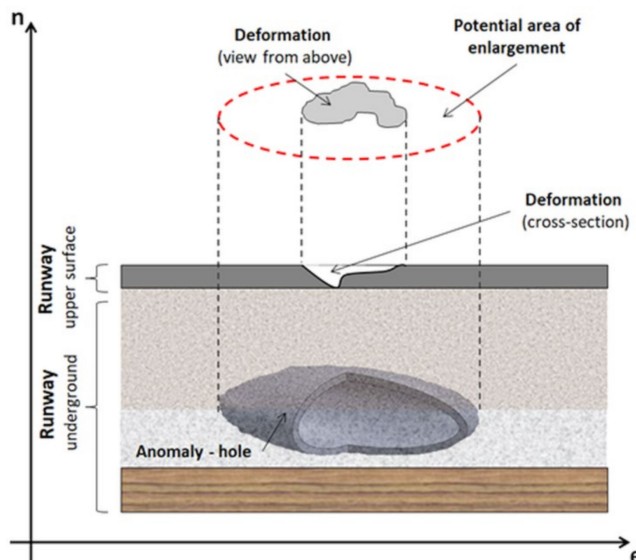

**Figure 18.** Schematic display of locational connection of the vertical deviation areas on the surface which are detected with geodetic measurements and anomalies in the lower structure detected with the FWD measurements.

## 4. Discussion

Runways are a part of airport infrastructure that are exposed to high loads; therefore, the monitoring of their condition is necessary. Deformed and poorly maintained runways directly affect safety. Lasting and vast renovations result in higher costs [51] and longer runway closures. Likewise, Liang et al. [52] stated that disturbances in air traffic have a large impact on the air industry. Disturbances that cause delays in air traffic also include undiscovered deformations on runways. These result in delays that are similar to those caused by aircraft damages. Moreover, Liang et al. [52] reported that there were 59 million min worth of delays in passenger air travel in 2016 at airports in the United States of America. The costs due to these delays increase every year and accounted for USD 74.24 per minute in 2019 [53]. Unlike larger and more congested airports, delays or even closures of less congested airports do not have extremely negative economic impacts on the airport owner or operator; however, late renovation of deformations causes an increase in their number and results in longer renovation time and higher costs for maintenance or refurbishment. These facts clearly show the need for efficient monitoring as a support for the updated and proper maintenance of all airports no matter the size or capacity.

Overall, deformation monitoring of pavement structures is a demanding process. The presented process model for maintenance and narrow process model of monitoring was developed in a way that suits smaller and less congested airports and enables multiple types of measurement data. As a result, our research recognizes the geodetic method to be suitable for inclusion in the monitoring model, and at the same time, it can be concluded that deformations or vertical deviation areas can be detected by usage of geodetic observations alone as also confirmed by Doler and Kovačič [16]. However, the reasons for their appearance cannot be determined and their enlargement cannot be forecasted. The results of the research and above-described facts can, therefore, allow to partly confirm this hypothesis. For the detailed consideration of deformations or vertical deviations which also includes the consideration of the reasons for their creation and allows the prediction of expansion of the deformations or vertical deviations, it is necessary to also include supplementary measurements with which the activities in the lower structure of the runway pavement surface can be researched. With the research, we also conclude that for the execution of the measurements, it is not necessary to close the airport; however, it is necessary to prepare an exact plan which assumes the phaseness of the execution of the measurements. With appropriate meteorological conditions, the measurements can be executed before the daily airport opening.

As described in Section 3.3. and shown in Figures 17 and 18, the special focus of our research was the fact that all measurements were in the same coordinate system, which is officially recognized in Slovenia. This provided for the interactive comparison of geodetic and FWD measurements, resulting in the conclusion that deformations or areas of vertical deviations on the upper surface and anomalies in the underground, causing larger deflections of the runway structure, locally coincide. These findings suggest a causal connection and indicate the existence of interactions between anomalies under the surface (cracks, not thick enough or poorly strengthened runway pavement, poorly built-in connection cables for the lighting of the runway, etc.) and deformations or vertical deviations arising on the runway's surface. Identifying the causes of formation of vertical deviations and deformations on the runway surface is not the subject of this research; however, establishment of these reasons is important, especially in terms of predictions for the formation of new deformations and enlargement of the existing ones as well as vertical deviations. Moreover, De Souza and De Almeida Filho [54] claimed that the anomalies already existing underground impact the formation and enlargement of deformations and vertical deviations. In summary, the authors' findings and conclusions during this study provide an estimation of deformation enlargement and further model development, which includes algorithms for the prediction of formation and enlargement of deformations and vertical deviations.

## 5. Conclusions

Airports are complicated structures; therefore, their management is very demanding. Efficient management requires many data about air facilities, activities, and surroundings. In aviation, management and maintenance receive much attention because well-maintained airport infrastructure directly correlates to the safety of air traffic. International organizations in aviation regulate normatively and direct maintenance works with recommendations. In terms of maintenance, larger airports are well regulated, while smaller ones are not; they do only the necessary work, i.e., only normative work and pay less attention to the development of systems that serve as maintenance support. Poor or even inappropriate monitoring is detected in smaller and less burdened airports that do heed suitable monitoring or do not have enough resources for its execution.

The goal of this research was to study the existing monitoring at smaller and less congested airports and to develop and test the monitoring model, which allows for advanced detection of deformations and areas of vertical deviations. The model was elaborated for the needs of monitoring at the Edvard Rusjan Airport in Maribor. First of all, the model development and monitoring of the results' relevance and, consequently, the model's calibration were long-lasting because deformation of pavement structures is a relatively slow process; in our case, it was also difficult from an organizational–logistical point of view, since the airport has a specific environment where movement is very restricted. Nevertheless, our study proved that the usage of methods, where a physical presence on the runway is needed despite problems associated with restricted movement, can also be appropriate for incorporation into the monitoring model. Simultaneously, the results of this approach will arouse greater confidence among decision-makers because they can be present at all times and observe the entire process of monitoring, starting with data capturing.

The list and description of the processes in this article provide a review of the procedures for monitoring and maintenance. The model results are highly valued in decision making about airport maintenance and, above all, about runway maintenance. Equally, the model is useful for monitoring other pavement structures, such as park areas, asphalt playing surfaces, and other surfaces, that are subject to deformation.

**Author Contributions:** Conceptualization: D.D, D.Ž. and B.K.; methodology: D.D.; software: D.D.; validation: D.D. and B.K.; formal analysis: D.D. and B.K.; investigation: D.D., B.K. and D.Ž.; resources: B.K.; data curation: D.D.; writing—original draft preparation: D.D., B.K. and D.Ž.; writing—review and editing: D.D. and B.K.; visualization: D.D.; supervision: B.K.; project administration: D.D. and B.K.; All authors have read and agreed to the published version of the manuscript.

**Funding:** This research received no external funding.

**Acknowledgments:** We thank the airport staff for allowing us to conduct the research.

**Conflicts of Interest:** The authors declare no conflict of interest.

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
