# Peer review of "Prototype of the Runway Monitoring Process at Smaller Airports: Edvard Rusjan Airport Maribor"

_processes, doi:10.3390/pr8121689_

Round 1

Reviewer 1 Report

- In the Abstract, Introduction or Materials and Methods nowhere explicitly mentioned working hypothesis. Likewise, the Discussion does not explicitly state that this working hypothesis was accepted or rejected.   - The abstract should be a total of about 200 words maximum. The abstract in the paper has more than 200 words.   - It is not entirely clear what the novelties in this article are, since some already published papers by the same authors have dealt with the same subject of research (methods, process, figures, etc.).      

Author Response

We edited the article according to your comments. We hope we have answered all your questions and comments. We are enclosing an edited article (track changes).

Response to Reviewer 1

Point 1: In the Abstract, Introduction or Materials and Methods nowhere explicitly mentioned working hypothesis. Likewise, the Discussion does not explicitly state that this working hypothesis was accepted or rejected.   - The abstract should be a total of about 200 words maximum. The abstract in the paper has more than 200 words.   - It is not entirely clear what the novelties in this article are, since some already published papers by the same authors have dealt with the same subject of research (methods, process, figures, etc.).

Response 1: We have considered all reviewer 1´s comments. In the text, changes are visible based on the included function “Track changes.

Reviewer 2 Report

Nice to read about a dedicated procedure with a clear goal. Also nice to read the last sentences of the paper, telling the described method is highly valued.

Some comments to the content:

On the measurement part (both geodetic and supplementary): Nothing is written on the required accuracy/quality of the measurements, and the obtained accuracy/quality/reliability. This kind of aspects is expected to be included for an academic paper involving measurements of any kind. Please add. The requirements to the measurements could possibly be connected (derived from?) the prescribed requirements to the runway given in lines 302 - 306.

The measurement trailer seems to be equipped with both GNSS positioning capabilities, and total station/prism. Please add information on which of the positioning equipment are used when, and how matching of the positions measured are ensured.

Very little is written on alternative methods to the one presented. Is seems obvious that some kind of laser scanner, terrestrial or air born, could be an alternative method for measuring deviations. Arguments for choosing the "traditional surveying procedure" should be added. Some statements on the appropriateness of the described methods are "hidden" in lines 273 - 276. Should be moved forward and further explained.

It is written several times (although not formulated in the research questions) that the goal is as little close-down time as possible for the airport. Then it is expected that some numbers on the duration of each measurement campaign and the expected needed frequency/repetition rate for the campaigns are presented. The only statement of this frequency is found in Picture 3 and 4, where is is written that the "... measures need to be repeated after a certain period" - a bit too vague.

Line 197: Please add a reference for the Hannover method

Line 199: Pleas add a reference to the Moor-Penrose method

Picture 1: Please add at least airport names to the Slovenia map. Seems also to be a mismatch between the size/content of the square for the Maribour Airport in the overview map compared to the detailed picture of the airport.

Picture 2: Nice to see BPMN diagrams. However, the name of the pool for connecting the bottom five lanes have misleading name according to BPMN conventions. Should be changed to "Maribour Edvard Rusjan Airport administration" or something similar. The data bases in the upper pool (The CB-LetMB data bases) seems according to the diagram to be resources outside the airport administration. Please also add a lane name her, indicating the actor/owner of the resources.

Picture 16: Not easy to interpret the message on comparism between the two methods. Should be made more clear.

It is consequently used the word "Refence" in front of the reference numbers, e.g. "....Reference [2]....". The word Reference should be removed throughout the paper.

Author Response

We edited the article according to your comments. We hope we have answered all your questions and comments. We are enclosing an edited article (track changes).

Response to Reviewer 2 

Point 1: On the measurement part (both geodetic and supplementary): Nothing is written on the required accuracy/quality of the measurements, and the obtained accuracy/quality/reliability. This kind of aspects is expected to be included for an academic paper involving measurements of any kind. Please add. The requirements to the measurements could possibly be connected (derived from?) the prescribed requirements to the runway given in lines 302 - 306.

Response 1: Chapter 2.2.4 is added.

Point 2: The measurement trailer seems to be equipped with both GNSS positioning capabilities, and total station/prism. Please add information on which of the positioning equipment are used when, and how matching of the positions measured are ensured.

Response 2: For the research, we have put the mobile GNSS receiver along with the measuring trailer and the measuring prism. Due to the inadequate frequency of recording the measurements and inadequate accuracy to determine altitude component, we have excluded the GNSS measurements from the follow-up consideration. We must warn that the GNSS method was used when establishing the zero geodetic net.

Point 3: Very little is written on alternative methods to the one presented. Is seems obvious that some kind of laser scanner, terrestrial or air born, could be an alternative method for measuring deviations. Arguments for choosing the "traditional surveying procedure" should be added. Some statements on the appropriateness of the described methods are "hidden" in lines 273 - 276. Should be moved forward and further explained.

Response 3: In the text between the lines 273–276 we explain that along with several methods of the FWD analysis measurements which are named in the chapter between the lines 266–268, we have chosen the approach which is based on the assessment of the structural capacity of the pavement structure determined based on the main deflexion – D0. For better understanding, we have added the abbreviation »FWD«. Changed text is worded as follows:

»The FWD measurement analysis can be executed on the basis of the calculation of the elasticity modules, on the basis of the structural capacity of the pavement’s structure according to the main deflection, or on the basis of the calculation of the deflection indices. 

The FWD measurement analysis in our research was executed by the approach based on the evaluation of structural capacity of the pavement structure, determined in respect to the deflection D0.«

Point 4: It is written several times (although not formulated in the research questions) that the goal is as little close-down time as possible for the airport. Then it is expected that some numbers on the duration of each measurement campaign and the expected needed frequency/repetition rate for the campaigns are presented. The only statement of this frequency is found in Picture 3 and 4, where is written that the "... measures need to be repeated after a certain period" - a bit too vague.

Response 4: In the article, we have added descriptions with which we provide the conclusions and explain the execution time component. In chapter 2.2.2., the following text I added:

»For the execution of the geodetic measurements on the measuring area, we needed 2 hours and 30 minutes in the first phase. In the second phase, we have executed three time dimensions. For each time dimension, we needed 30 minutes.«.

In chapter 2.2.3., the following text is added:

»For the execution of the FWD measurements on the measuring area, we needed 1 hour and 10 minutes.«.

In the discussion, the following text is added:

»With the research, we also conclude that for the execution of the measurements it is not necessary to close the airport; however, it is necessary to prepare an exact plan which assumes the phaseness of the execution of the measurements. With appropriate meteorological conditions, the measurements can be executed before the daily airport opening.«.

Point 5: Line 197: Please add a reference for the Hannover method.

Response 5: The reference is added [42].

Point 6: Line 199: Pleas add a reference to the Moor-Penrose method.

Response 6: The reference is added [43].

Point 7: Picture 1: Please add at least airport names to the Slovenia map. Seems also to be a mismatch between the size/content of the square for the Maribour Airport in the overview map compared to the detailed picture of the airport.

Response 7: Figure 1 was adequately corrected and the names of international airports in Slovenia were added.

Point 8: Picture 2: Nice to see BPMN diagrams. However, the name of the pool for connecting the bottom five lanes have misleading name according to BPMN conventions. Should be changed to "Maribour Edvard Rusjan Airport administration" or something similar. The data bases in the upper pool (The CB-LetMB data bases) seems according to the diagram to be resources outside the airport administration. Please also add a lane name her, indicating the actor/owner of the resources.

Response 8: BPMN diagram in Figure 2 is corrected.

Point 9: Picture 16: Not easy to interpret the message on comparism between the two methods. Should be made more clear.

Response 9: Figure 16 does not show the comparison but the combination of the geodetic and FWD method. Figure 16 was fulfilled, and Figure 17 was added. In Chapter 3.2, the following text with the detailed description of Figure 16 is added:

»In Figure 16, three areas with red, black and blue circle parts are marked with numbers 1, 2 and 3 which coincide with the areas of larger vertical deviations detected with the use of geodetic measurements and areas where with the FWD measurements bigger deflection values were measured. Bigger deflection values are in Figure 16, marked on the graph, with squares and numbers 1, 2 in 3.«.

Above Figure 17, the following text is added:

»The comparison of the results of both measurements suggest the occurrence of a causal connection between the perceived vertical deviations that occur on the runaway surface and anomalies in the runway underground. As a result, this finding provides further model development, which also enables the forecast of the appearance of vertical deviations. Figure 17 shows a schematic display of locational connection of the vertical deviations areas on the surface which are detected with geodetic measurements and anomalies in the lower structure, detected with the FWD measurements.«.

Point 10: It is consequently used the word "Refence" in front of the reference numbers, e.g. "....Reference [2]....". The word Reference should be removed throughout the paper.

Response 10: The word ˝Reference˝ is deleted or substituted with the adequate text.

Reviewer 3 Report

There is a research on prototype of the runway monitoring process at smaller airports. This paper is very interesting article and provide some insights on this area. I think that the present writing would be good to publish.

Author Response

Response to Reviewer 3

We edited the article based on the comments of other reviewers. We are enclosing an edited article (track changes).

Round 2

Reviewer 1 Report

Well done

Reviewer 2 Report

Proper handling of the comments from first review.